# Characteristics of Sedimentary Organic Matter in Tidal Estuaries: A Case Study from the Minjiang River Estuary

Shuilan Wu [1,2], Shuqin Tao [1,3,*], Xiang Ye [1,3,*], Aijun Wang [1,3,4], Zitong Liu [1,5], Chang Ran [1,3], Haoshen Liang [1], Haiqi Li [1,6], Yuxin Yang [1], Wangze Zhang [1,7] and James T. Liu [8]

1    Third Institute of Oceanography, Ministry of Natural Resources (MNR), Xiamen 361005, China; 15107874031@163.com (S.W.)
2    School of Marine Sciences, Nanjing University of Information Science & Technology, Nanjing 210044, China
3    Fujian Provincial Key Laboratory of Marine Physical and Geological Processes, Xiamen 361005, China
4    Southern Marine Science and Engineering Guangdong Laboratory (Zhuhai), Zhuhai 519082, China
5    Institute of Marine Sciences, Shantou University, Shantou 515063, China
6    College of Marine Sciences, Shanghai Ocean University, Shanghai 200090, China
7    School of Advanced Manufacturing, Fuzhou University, Fuzhou 350108, China
8    Department of Oceanography, Sun Yat-sen University, Kaohsiung 80424, Taiwan; james@mail.nsysu.edu.tw
*    Correspondence: taoshuqin@tio.org.cn (S.T.); yexiang@tio.org.cn (X.Y.); Tel./Fax: +86-592-2195-056 (S.T.); +86-592-2195-112 (X.Y.)

**Abstract:** As one of the main interfaces of the Earth system, estuaries show the strongest land–sea interaction in the carbon cycle, which links terrestrial ecosystems to the marginal sea. Furthermore, estuaries are considered as one of the most active intermediate reservoirs for both terrestrial and marine matter due to complex hydrodynamic processes regulated by the river runoff, wave and tide. Processing of organic matter (OM) in tidal estuaries modifies its transfer and transformation from the river to the sea, so studies of on the source and distributions of estuarine OM can help us understand the behavior of production, exchange, transport and burial of diverse OM within this transition zone before entering the marginal sea. In this paper, we took the Minjiang River Estuary (MRE) as a typical system in which there is strong influence of the tide. The source, composition and spatial distribution of OM in surface sediments of MRE were deciphered based on multiple organic geochemical properties for source-specific biomarkers (*n*-alkanes, *n*-alkanols, sterols) and bulk OM. Results show that sedimentary organic components were negatively correlated with sediment grain size, which indicates fine particles such as silt and clay are the major carriers of the OM signals in tidal estuaries. Source-specific biomarker proxies indicate that in terms of source diversity the sedimentary OM in the MRE shows mixed signals of terrestrial and marine sources, and the proportion of terrestrial OM decreases with the increase in distance from the land. The fractional contributions of OM from the riverine (i.e., terrestrial), marine and deltaic sources were quantitatively estimated using a Monte Carlo (MC) three-end-member mixing model based on C/N and $\delta^{13}C$ values, and the average contributions of the three sources are $40 \pm 10\%$, $48 \pm 10\%$ and $12 \pm 4\%$, respectively, with little contribution from deltaic sources. The dispersion of sedimentary OM from different sources in the MRE is primarily controlled by the depositional environment determined by dynamic conditions and tidal processes play a significant role in the redistribution of sedimentary OM dispersion patterns. Compared with other large estuaries in southeast China, the OM accumulation contribution in the tide dominated small and medium-sized estuaries such as the MRE which is largely dependent on riverine and marine deliveries. The MRE has a high potential for both terrestrial and marine organic carbon (OC) burial, with an accumulation rate of $3.39 \pm 1.83$ mg cm$^{-2}$ yr$^{-1}$ for terrestrial OC, and an accumulation rate of $3.18 \pm 0.68$ mg cm$^{-2}$ yr$^{-1}$ for marine OC in muddy sediment, making it an important contributor to the sedimentary carbon sink of the marginal sea.

**Keywords:** organic matter sources; biomarkers; stable carbon isotopes; end-member mixing model; Minjiang River Estuary

## 1. Introduction

Estuary area is a key point in the global organic carbon cycle, accounting for about 40–50% of global marine sediment organic carbon sequestration [1,2]. Meanwhile, the characteristics of "high primary productivity and high deposition rate" in estuaries and sub-aqueous deltas make these systems often become marine organic carbon reservoirs [1,3,4]. Although estuaries are commonly a source of strong $CO_2$ emissions in terms of air–sea $CO_2$ fluxes [5], they also serve as important burial centers of terrestrial organic carbon (OC). Each year, rivers transport approximately 0.85 Pg C (Pg = $10^{15}$ g) from the continents to estuaries [6]. The carbon cycling process in tidal estuaries is exceptionally complex, influenced by a variety of dynamic factors, such as runoff, tides and waves. Further, secondary interfaces such as salt–freshwater fronts, ebb and flow transition interface and diffusion interfaces of diluted water, and special interfaces such as maximum turbidity zones (TMZ) act as "filters" for riverine materials transited to the ocean [7–9]. According to the studies, TMZ were found to increase the retention time of particulate organic matter (OM) in the estuary. However, frequent resuspension increased the exposure time of OM to the aerobic environment, thereby intensifying the degradation process [10–12]. This was particularly evident in estuaries with strong tides, in which the dynamic salt–freshwater exchange and particle settling–resuspension transitions continuously intensified the extensive degradation of OM [5,12,13]. However, when studying the carbon cycle in the waters in continental margins, researchers commonly consider the input fluxes and the endmember characteristics of OM at the hydrological observation station closest to the estuary when determining the boundary condition of the sea–land interface. Due to the limitations in spatial resolution, the behavior and process of OM within the estuary are often neglected, leading to uncertainties in constraining the terrestrial end member of the carbon cycle in the adjacent marginal seas [14,15]. In river-dominated systems, most riverine particles and associated organic matter are deposited in estuaries. High levels of land-derived nutrient inputs support high primary production through the photosynthesis process within the estuaries, whereas highly microbial activities and diversity at this region enhance OM mineralization and induce selective degradation [14,16,17]. Sedimentary OM buried in the estuarine is an important dataset for reconstructing the environmental changes. By analyzing the source and composition of sedimentary OM in estuaries, we can evaluate the processes of OM generation, transportation and burial, and the behavior and processes of organic carbon from different sources entering estuary and source-sink patterns in this area.

The buried efficiencies and pattern of organic carbon in different types of estuarine systems are diverse [1]. The majority of existing studies concentrate on the organic carbon cycling in estuarine system under the influence of large rivers [3,18–20] and mid-small mountainous river-narrow continental shelf systems [21–24]; but, there are not enough studies on organic carbon cycling processes in mid-small mountainous river-wide continental shelf systems [21,25,26]. Compared to the vast sedimentary systems found in large river basins and estuaries, mid-small mountainous rivers with short and steep flow have smaller drainage areas and smaller nearshore sedimentary systems, resulting in faster lateral transport into the sea. As a result, the sedimentary OM in estuaries is more responsive to changes in the climate and environment within the watershed [1,23,27], providing a better reflection of terrestrial source signals and information on changes in the transport of terrigenous material into the sea [21,24,27]. The processing of OM in tidal estuaries is disparate compared with that in mobile deltaic shelf sediments such as the Yangtze and Amazon–Guianas estuaries [28,29]. Especially, tidal estuaries are characterized by long residence times of water and particles due to cycles of deposition and resuspension and recurrent change oxic–anoxic oscillations [30], which result in extensive modification of organic matter in the inner estuary [12,31]. Thus, it is essential to fill in the knowledge gap regarding detailed distinction in the sedimentary OM distribution patterns and diversity in this tidally regulated deltaic portion of source-to-sink sedimentary systems.

The Minjiang River is the largest mountainous river along the southeast coast of China, which pours into the East China Sea to form a medium sized mountainous river mouth

system, with obvious deltaic deposits zonation [32]. The Minjiang River Estuary (MRE) is a strong tidal estuary with an average tidal range of 4.46 m and a high tidal current velocity, which is characterized by abundant sediment supply and strong tidally induced hydrodynamic force [33]. The Minjiang River is significantly affected by extreme events due to tropical cyclones generated in the northwest Pacific Ocean and the South China Sea [34–36]. In this paper, we will use source-specific biomarkers, elemental and stable isotope data ($\delta^{13}$C) of bulk OM extracted from surface sediments collected from MRE to decipher the OM compositions and their distribution pathways in the strong tidal estuary.

## 2. Research Areas and Methods

### 2.1. Study Area

Minjiang River is the largest mountainous river along the southeast coast of China, with a total length of 577 km and a drainage area of 60,992 km$^2$. The MRE is divided into the North Branch (Changmen Channel, CMC) and the South Branch (Meihua Channel, MHC) below Tingjiang, and into several secondary branches in the CMC; the Chuanshi Channel (CSC) is the main channel of the MRE (Figure 1), which eventually pours into the East China Sea. The MRE belongs to subtropical marine monsoon climate, with warm and humid throughout the year and abundant rainfall, and the annual average runoff is 605.5 × 10$^8$ m$^3$ and the annual average sediment discharge is 750 × 10$^4$ t; the flood season (March–August) accounts for about 76% of the annual runoff and 92% of the annual sediment discharge. The mean tidal range is 4.46 m at Meihua station, the mean flood tidal duration is 5 h and 31 min, the mean ebb tidal duration is 6 h and 54 min and the multi-year average wave height is 1.1 m, which belongs to the macrotidal estuary. Under the combination of special geological structure, topography, sediment flux and estuarine dynamics, a large number of underwater deltas have developed in the MRE area [32].

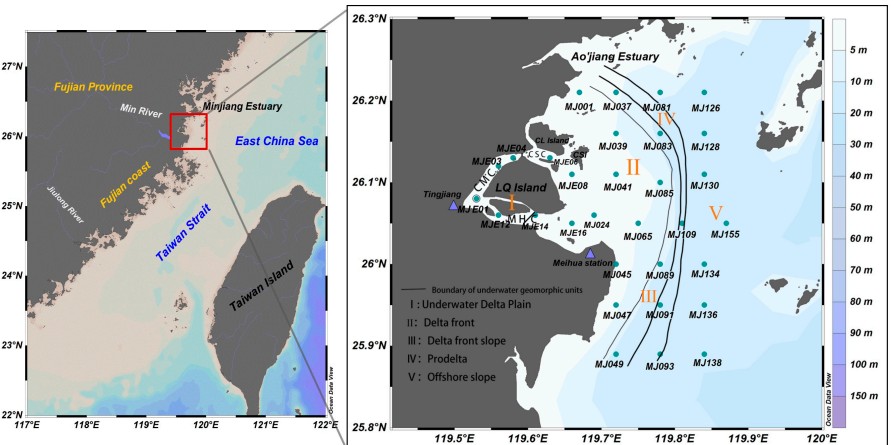

**Figure 1.** Schematic diagram of MRE area and sampling stations. LQ Island: Langqi Island, CL Island: Culu Island, CSI: Chuanshi Island, CMC: Changmen Channel, MHC: Meihua Channel. The deltaic zonation is modified from Chen et al. (1999) [37].

### 2.2. Sample Collections and Analyses

Surface (1 cm) sediment samples (*n* = 31) in the MRE were collected in August 2019 (the sites are shown in Figure 1). Sediment samples were retrieved by grab sampler and stored at −20 °C prior to analysis. The on-site environmental parameters of water temperature, salinity and turbidity at 31 sites were collected. In the laboratory, sediment samples were lyophilized using an Eyela FDU-2100 freeze dryer (Tokyo Rikakikai, Shanghai, China), then homogeneously grounded into powder with a mortar and stored in a desiccator until analysis.

### 2.3. Sediment Grain Size Analysis

For grain size analyses, 5 mL of 10% $H_2O_2$ were added to ~0.2 g of surface sediment sample at the room temperature for 24 h to oxidize the OM. Then 1 mL of 3 mol/L HCl was added to the sample at the room temperature for 24 h to remove calcium carbonate. After centrifuging and decanting several times with Milli-Q water to wash out the acid until neutral, 0.5% $Na(PO_3)_6$ solution was added to the sample at room temperature for 24 h to disperse the sample fully. The processed samples were analyzed using a laser particle sizer (Mastersizer 2000, Malvern, UK) and the results were exported using the software that comes with the instrument. The average particle size of the sediment was calculated using the method of moments [38] and the sediment type was classified using the Shepard classification system [39].

### 2.4. Bulk OM Characteristics Analysis

For bulk element content analyses, after freeze drying and grinding, 1 mol/L HCl solution was added to a certain amount of ground sample at room temperature for 24 h to remove the inorganic portion (mainly in the form of carbonate), and then acidified samples were rinsed, centrifuged and decanted 5–7 times with Milli-Q water until neutral. Afterwards, well-treated samples were then dried at 60 °C and carefully crimp-sealed in tin capsules for element-content analyses. Ground samples were taken before and after acidification to determine the carbon and nitrogen content of the sediment samples using an elemental analyzer (Elementar Vario ELIII, ELEMENTOR, Berlin, Germany) to obtain the contents of total organic carbon (TOC) and total nitrogen (TN) for the sediment through mass conversion. A certain amount of the acidified sample was weighed for carbon isotope ratio ($\delta^{13}C$) analysis using a stable isotope ratio mass spectrometer (Thermo MAT253 IRMS, Thermo, Tokyo, Japan). The precision for TOC and TN content, $\delta^{13}C$ is better than ±0.2% and ±0.2‰, respectively.

### 2.5. Biomarker Analysis

The detailed methods for extraction, purification and isolation of target biomarkers have been provided and modified based on Tao et al. (2022) [25]. About 5 g of unacidified ground sample was weighed into a sample vial and biomarker internal standards ($n$-$C_{24}D_{50}$, $n$-$C_{19}$ alkanol, and $n$-$C_{19}$ fatty acid) were added. The total lipids in the sample were extracted by extraction with a mixture of dichloromethane (DCM) and methanol (MeOH) (3:1, $v/v$) 4 times via ultrasonication for 20 min and the solvent was blown dry with high purity nitrogen. To the sample vial, 5 mL of 6% ($w/w$) KOH methanolic solution and a few drops of Milli-Q water were added and placed at constant temperature for overnight saponification reaction. Then, the solution was extracted with hexane liquid–liquid three times and the supernatant was combined to obtain the neutral lipid fraction. Afterward, a "neutral" fraction and an "acid" fraction were back-extracted from the hydrolyzed solution separately after adjusting the pH of solution. The "neutral" fraction was further separated by $SiO_2$ column chromatography into two fractions: the non-polar fraction eluted by 8 mL hexane containing n-alkanes and the polar fraction eluted by 12 mL DCM/MeOH (95:5, $v/v$) containing plant wax $n$-alkanols and sterols. N-Alkanes were concentrated under $N_2$ to 60 μL with iso-octane before measurements. N-Alkanols and sterols were derivatized using N,O-bis (trimethylsilyl) trifluoroacetamide (BSTFA) at 70 °C for 1 h before measurements.

The chemically purified alkanes, alkanols and sterols were analyzed by high performance gas chromatography (Agilent Technologies 7890B GC-FID, Santa Clara, CA, USA). Identification of different biomarkers by comparing the peak retention times of samples and $C_8$–$C_{40}$ alkanes, $C_{16}$–$C_{30}$ straight chain alkanols, phyto-alkanols and 7 sterol mixtures (coprostanol, cholesterol, brassicasterol stigmasterol, sitosterol and dinosterol) for qualitative analysis. The internal standard method was used to conduct quantitative analysis of the different biomarker components. The relative deviation of the biomarker concentration measurement data was <15%.

Based on the results of GC measure to *n*-alkanes, *n*-alkanols and sterols, the biomarker parameter TMBR is calculated to identify the source of OM in surface sediments of MRE. The TMBR (terrestrial and marine biomarker ratios) index, an indicator of terrestrial plants and aquatic phytoplankton biomarkers ratios, was calculated with the formula [40]:

$$\text{TMBR} = \frac{n-C_{27+29+31}alkanes}{n-C_{27+29+31}alkanes + A + B + C} \tag{1}$$

or

$$\text{TMBRol} = \frac{n-C_{28+30+32}alcohols}{n-C_{28+30+32}alcohols + A + B + C} \tag{2}$$

where *A*, *B*, *C* represent $C_{37}$ alkenones, brassicasterol and dinosterol, respectively. TMBR index ranges from 0 to 1; 0 means that all the sedimentary OM comes from marine and TMBR = 1 means that all the sedimentary OM comes from terrene.

### 2.6. Statistical Analysis

The principal component analysis (PCA) is a multivariate statistical analysis method, using fewer representative principal components to represent most of the information [41]. In this paper, PCA was performed using SPSS software to identify the most representative components and variables of sedimentary OM in the MRE, aiming to assess statistical differences in the spatial distributions of sedimentary OM. The raw data were first standardized to derive the correlation coefficients between the indicators. Then the standardized data were subjected to principal component analysis to identify the principal components based on their contribution rates and calculate the loading matrix of the principal components. The main influencing factors on the spatial distributions of sedimentary OM of the MRE can be determined based on the load matrix analysis.

### 2.7. Model for Quantitative Estimation of Sedimentary OM Sources

The higher the C/N and the more negative the $\delta^{13}$C, the higher the terrestrial source OM input. In order to quantitatively assess the relative contribution of sedimentary OM from different sources of the MRE in this paper, C/N ratio and $\delta^{13}$C coupled three end member Monte Carlo (MC) model are used to calculate the burial contribution of marine source ($f_M$), riverine (i.e., terrestrial) source ($f_R$) and delta source organic matter ($f_D$, which is mainly the OM source of salt marsh in estuary) using python program; the three end member model formula is as follows:

$$\begin{aligned} \delta^{13}C_{sample} &= \delta^{13}C_M \times f_M + \delta^{13}C_R \times f_R + \delta^{13}C_D \times f_D \\ C/N_{sample} &= C/N_M \times f_M + C/N_R \times f_R + C/N_D \times f_D \\ f_M + f_R + f_D &= 1 \end{aligned} \tag{3}$$

The $\delta^{13}C_M$, $\delta^{13}C_R$ and $\delta^{13}C_D$ represent the $\delta^{13}$C end member values of marine, river and delta sources, respectively, and the $C/N_M$, $C/N_R$ and $C/N_D$ represent the C/N end member values of marine, river and delta sources, respectively.

## 3. Results

### 3.1. Water Mass Characteristics

The measured water mass properties during the investigation are illustrated in Figure 2 (the weighted-average method is used to describe spatial variability of all parameters in this article). The temperatures in the surface and bottom water varied spatially between 27.4–29.8 °C and 24.5–29.8 °C, respectively. Generally, the water temperature decreases gradually from the river mouth to the offshore slope (Figure 2a,b). The salinity of the surface and bottom water varied spatially from 0.05–33.55 and 0.55–33.93, respectively, of which the spatial pattern generally increases with distance from the river mouth (Figure 2c,d). The spatial distributions of surface and bottom water turbidity in the MRE varied between 1 and 418 FTU and 5 and 680 FTU, respectively, with higher turbidity near the water channel

and the delta front. Notably, the highest turbidity values were located at the extension of the Meihua Channel outlet, which might be associated with the phenomenon of turbidity maximum zone (Figure 2e,f). The spatial distribution of dissolved oxygen (DO%) in both surface and bottom waters varied between 74.1 and 100% and 66.2 and 105.7%, respectively. Low DO saturations were observed in both surface and bottom waters at the channels, while high DO saturations were found in surface water and low DO saturations in bottom water at the north and south of the prodelta and offshore slope (Figure 2g,h). The distribution characteristics of all water properties in the MRE reflect the dispersal of the fresh river effluent to the sea in summer during high temperature, low salinity and high turbidity, which was blocked by the low temperature, high salinity and low turbidity coastal water on the west side of the Taiwan Strait in summer. The river water dispersal was impeded in the delta front of the estuary. The spatial patterns of these water properties seem to suggest the MHC was the dominant outlet for the dispersal of the Minjiang River effluent.

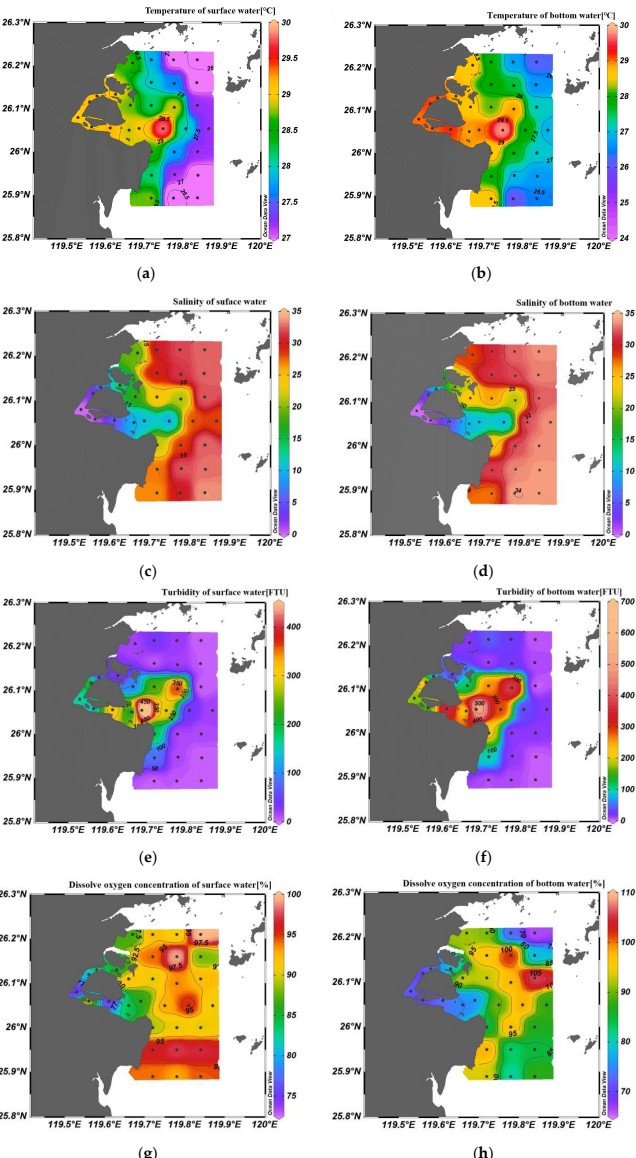

**Figure 2.** Spatial variations of environmental parameters at surface and bottom layers at the MRE: (**a**) surface temperature; (**b**) bottom temperature; (**c**) surface salinity; (**d**) bottom salinity; (**e**) surface turbidity; (**f**) bottom turbidity; (**g**) surface dissolve oxygen saturation; (**h**) bottom dissolve oxygen saturations.

### 3.2. Bulk Characteristics

The spatial distribution of the grain size composition of the surface sediments in the MRE show that the sand content ranged from 0.6% to 100%, with an average of 60.9%. The content of sand generally decreased with distance from the river mouth, with higher proportions at the channels, the shoal of the delta front at the north of Meihua station and the middle part of offshore slope to the east (Figure 3a). The silt and clay contents ranged from 0 to 68.4% and 0 to 32.6%, with an average of 27.1% and 12.0%, respectively. The spatial distribution of the silt and clay content is opposite to that of the sand content, with a narrow belt of higher muddy sediments in the narrow region of the prodelta and offshore slope (Figure 3a–c). The mean grain size of surface sediments was 0.39~7.44 $\phi$ with an average of 3.81 $\phi$. The sediment mean grain size was coarser in the inlet channel and the shallow shoals in the north of Meihua Channel, but finer in the prodelta and its adjacent delta front areas. In the middle of the submerged offshore slope, the sediment mean grain size increased (Figure 3d).

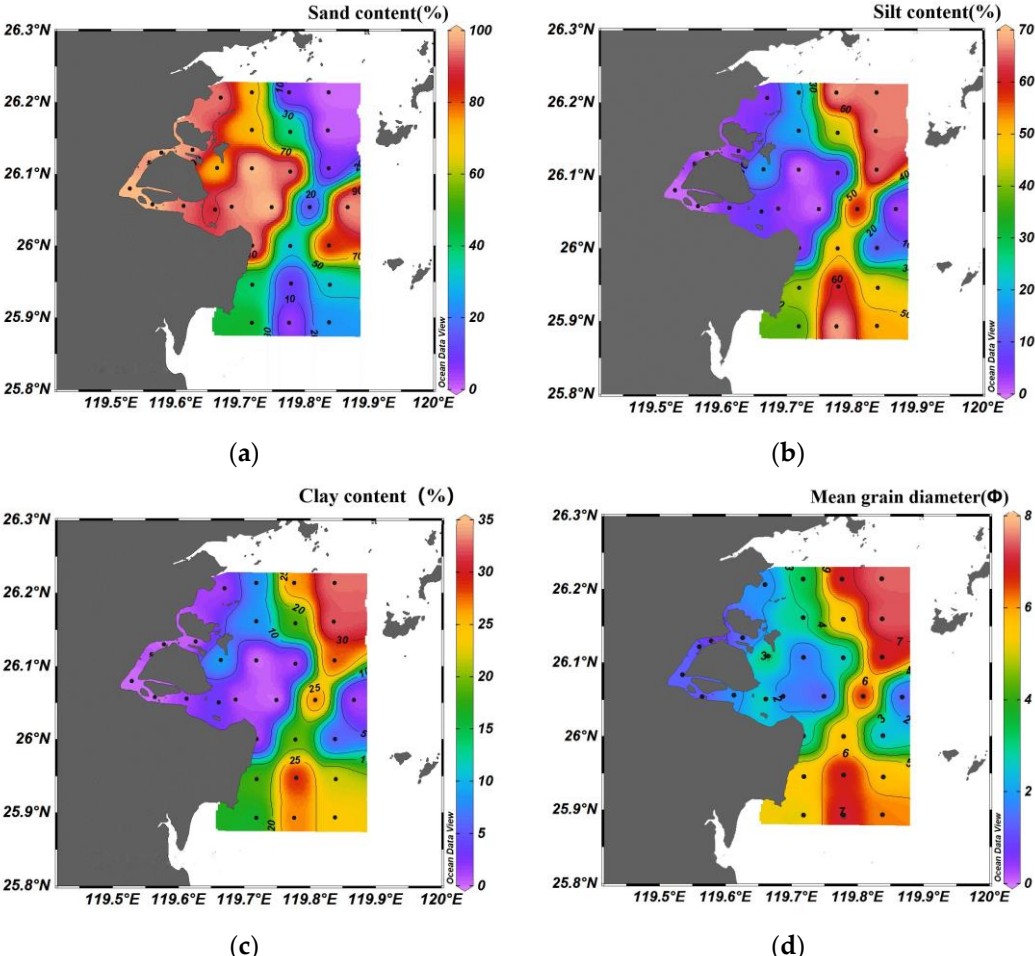

**Figure 3.** Spatial distribution of surface sediment composition and mean grain size: (**a**) sand content; (**b**) silt content; (**c**) clay content; (**d**) mean grain size ($\phi$).

There are five types of surface sediments in MRE. The coarse sand (S) is the most widely distributed, accounting for 45% of the total stations, most of which are distributed in the channel, the shallow sandy shoals at the north of the Meihua station and the offshore shelf area. Secondly, the clayey silt (YT) type accounts for 22% of the total stations and is distributed in the prodelta at the northeast and middle of the study area. Then silty sand (TS) and sandy silt (ST) type account for 16% and 10% of the total stations, respectively.

The type of sand–silt–clay (STY) is relatively small (6%), distributed in the southeast of the study area (Figure 4).

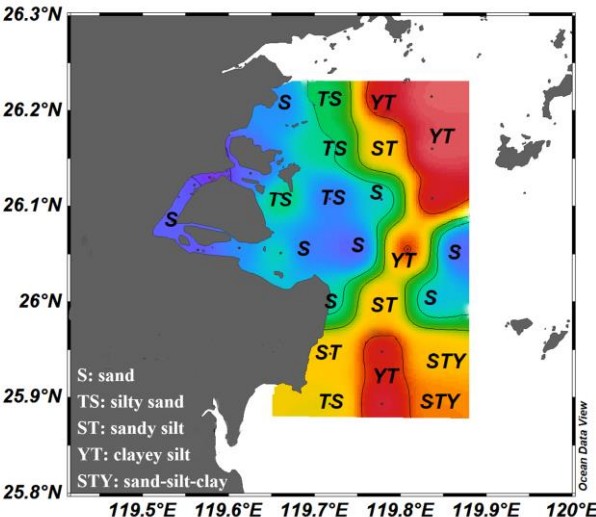

**Figure 4.** Spatial distribution of surface sediment classification.

The TOC content (TOC%) in the surface sediments of MRE varied spatially between 0.04 to 1.00%, with an average of 0.45%. The TN content (TN%) varied spatially between 0 and 0.14%, with an average value of 0.06% and the spatial patterns of the two are closely associated (Figure 5a,b). The content of TOC and TN is lower in channel, delta front and offshore shelf sandy sediment area, while high content of TOC and TN is observed in a "narrow muddy belt" located in the prodelta region, which has a similar distribution as the mean grain size of the sediment (Figure 3d). In other words, the content of TOC and TN was higher in the area with fine grained deposits and low in the area with coarse grained depositions. However, it is worth noting that the TOC and TN content at the outlet of the Meihua Channel, where sediment was predominantly composed of sandy particles, is significantly higher than those at the surrounding stations. Therefore, high TOC and TN contents in this area might reflect another OM accumulation mechanism at the MRE which is not associated with preservation regulated by the particle affinity of OM.

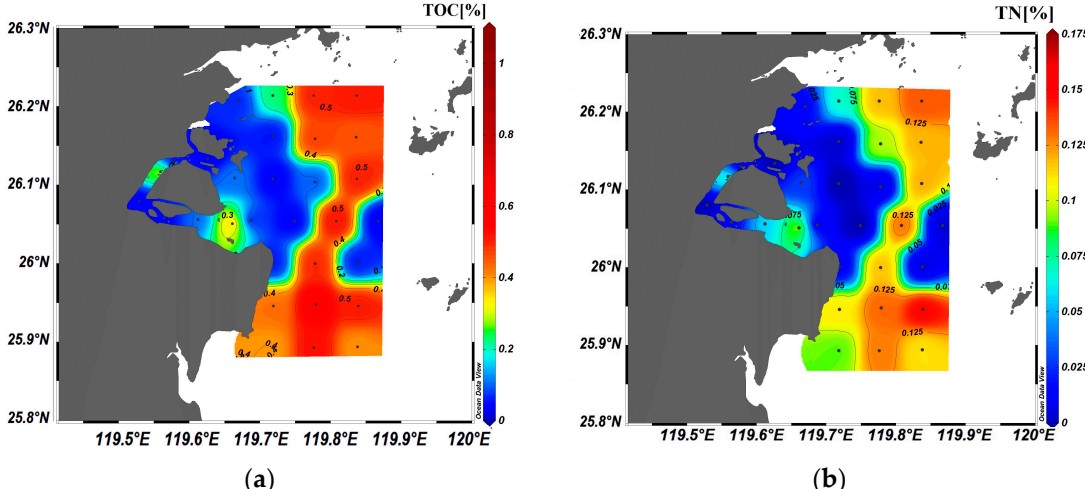

(**a**)  (**b**)

**Figure 5.** *Cont.*

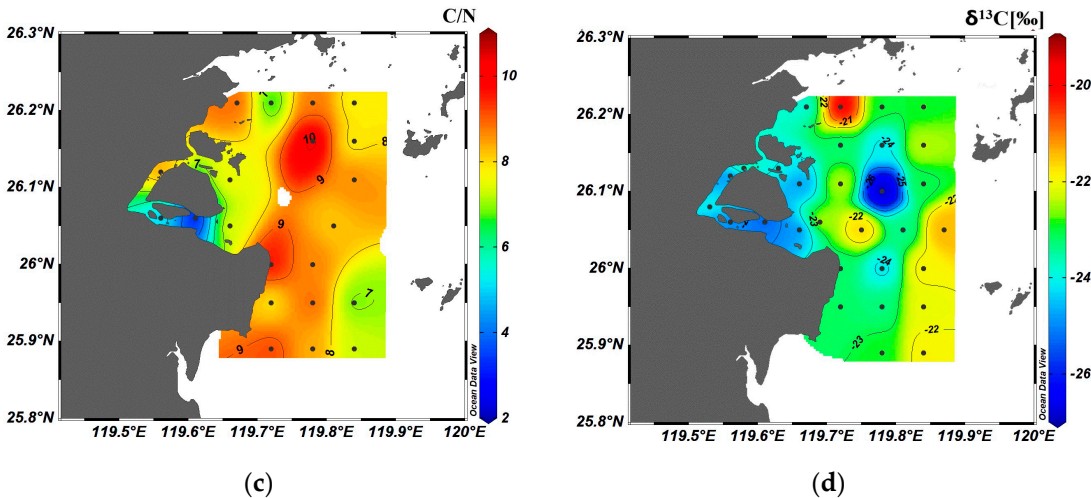

**Figure 5.** Spatial variations of (**a**) TOC content, (**b**) TN content, (**c**) C/N, (**d**) $\delta^{13}$C in surface sediments of MRE.

The C/N values of surface sedimentary OM varied from 2.97 to 10.24, with an average value of 7.89. The bulk OC $\delta^{13}$C values ranged from $-26.8‰$ to $-19.9‰$ with an average value of $-23.4‰$. The spatial patterns of C/N and $\delta^{13}$C values are shown in Figure 5c,d, respectively. C/N values are generally between 7 and 10, with lower ratios (2–6) observed at the Meihua Channel. The $\delta^{13}$C values show a narrow range of $-24--22‰$, with lower values observed at the underwater delta plain and the middle of delta front slope.

### 3.3. Content and Composition Characteristics of Source-Specific Molecular Biomarkers
3.3.1. Terrestrial Biomarkers

The epidermal waxy layers of the terrigenous vascular plants contain abundant $C_{27}$, $C_{29}$ and $C_{31}$ alkanes, so odd carbon number long chain $n$-alkanes ($C_{27}$, $C_{29}$, $C_{31}$) are often used as an indicator of terrestrial plant derived OM [42,43]. $n$-$C_{10}$–$n$-$C_{40}$ alkanes are detected in surface sediment samples of MRE, which displayed three types of distributions for all $n$-alkanes homologous. Most sites showed single peak distribution with high content in high carbon number alkanes and the contents of $C_{29}$, $C_{31}$ and $C_{33}$ are dominant. About one-third of sites showed double peak distributions with the high content both in high and low carbon numbers of alkanes. The higher content in low carbon number alkanes were found in a few sites, mainly concentrated on $C_{18}$, $C_{19}$ and $C_{20}$. The content of plant-wax derived $n$-alkanes in MRE surface sediments, expressed as the total content of $n$-$C_{27}$, $n$-$C_{29}$ and $n$-$C_{31}$ alkanes ($n$-$C_{27+29+31}$ alkanes), varied spatially from 4 to 1688 ng/g, with an average of 272 ng/g. The low contents were mainly distributed at the estuarine channel, delta front and the middle of offshore slope in the east of the study area and the high contents were mainly distributed at the prodelta (Figure 6a), of which of the spatial patterns were generally similar to the distributions of TOC and TN content (Figure 5a,b).

Even carbon long chain $n$-alcohols ($C_{28}$, $C_{30}$, $C_{32}$) were also mainly derived from terrestrial vascular plant wax, which can be used as an indicator for terrestrial OM tracers in estuaries [44–46]. The contents of plant wax derived $n$-alkanols in MRE surface sediments, expressed as the total content of $n$-$C_{28}$, $n$-$C_{30}$ and $n$-$C_{32}$ alkanols ($n$-$C_{28+30+32}$ alkanols), varied from 14 to 5113 ng/g, with an average of 853 ng/g. In addition, the spatial distribution was similar to that of plant wax derived $n$-$C_{27+29+31}$ alkanes (Figure 6a,b).

β-sitosterols and stigmasterols are widely present in terrestrial vegetation cells [47] and coprostanol are mainly derived from human sewage discharges [48,49]. The total content of three typical terrestrial sterols (coprostanol, sitosterol and stigmasterol) varied from 16 to 2842 ng/g, with an average of 765 ng/g, of which the spatial distribution (Figure 6c) is similar to those of other terrestrial biomarkers (Figure 6a,b). The high contents were distributed at the middle of the delta front slope and prodelta.

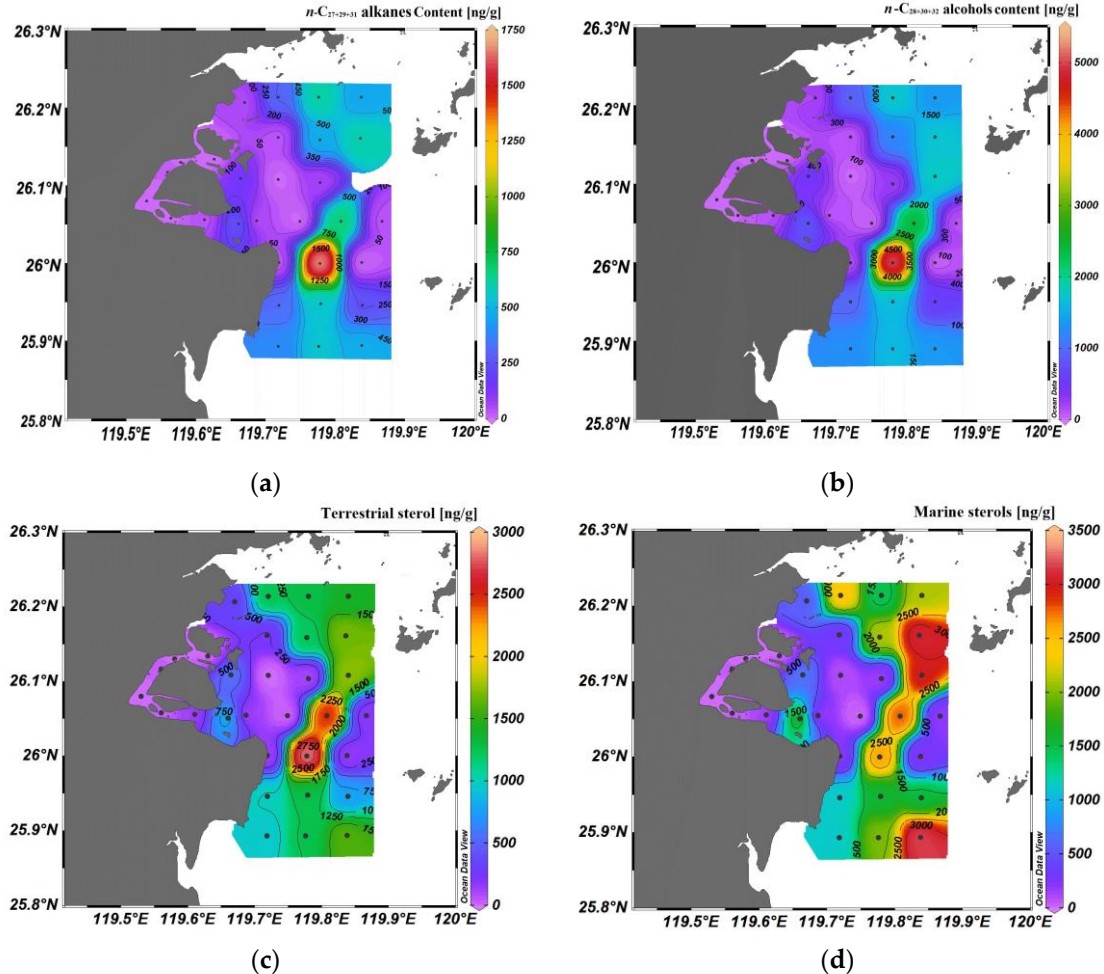

**Figure 6.** Spatial distributions of (**a**) *n*-C$_{27+29+31}$ alkanes, (**b**) *n*-C$_{28+30+32}$ alcohols, (**c**) terrestrial sterols and (**d**) marine sterols in surface sediments of MRE.

### 3.3.2. Marine Biomarkers

Brassicasterol and dinosterol are mainly derived from the aquatic phytoplankton diatoms and dinoflagellates, respectively, and cholesterol is usually considered as an indicator of zooplankton [50]. The total contents of three common marine biomass derived sterols (brassicasterol + dinosterol + cholesterol) varied from 13 to 3177 ng/g with an average of 1118 ng/g. Generally, the contents of marine biomass derived sterols were low in the delta front area and the sandy offshore shelf area and high in the delta front slope and prodelta region where sediment types were dominantly of clay silt and sand–silt–clay. Furthermore, there was relative high content at the outlet of the Meihua Channel than that of the other river mouth proximal sites (Figure 6d).

### 3.4. PCA Results

The spatial distributions of TOC, TN and source-specific biomarkers in surface sediments of MRE (Figures 5a,b and 6a–d) reveal that molecular compounds of different sources share similar characteristics. Specifically, the contents of all organic components were generally lower in coarse-grained sandy sediments but higher in fine-grained clayey sediments. The principal component load matrix obtained from PCA can reflect the correlation between individual indices and principal components. A higher absolute value of the load matrix coefficient indicates a stronger correlation between the principal component and the individual index. For PCA analysis, the dataset contained a total of 17 variables including bulk OM properties (TOC%, TN% and bulk OC $\delta^{13}$C), source-specific biomarkers contents, environmental physical parameters and co-ordinate information (Figure 7a). In general,

two principal components (PC 1 and 2) are responsible for 73.9% of the total variance in the data. PC 1 accounted for 62.2% of the total variance with high positive loadings of the content of TOC, TN, marine phytoplankton derived sterols (dinosterol + brassicasterol), terrestrial sterols and fine-particle related parameters (clay and silt). Coarse-particle related parameter (the sand content) is negatively loaded on PC 1. PC 2 only accounts for 11.7% of the total variance, with high positive loadings of terrestrial *n*-alkyl biomarkers content ($n$-$C_{27+29+31}$ alkanes and $n$-$C_{28+30+32}$ alkanols) and hydrological parameters such as water temperature and turbidity. Bulk OC $\delta^{13}C$ values and salinity are negatively loaded on PC 2 (Figure 7a). According to the PCA load matrix, PC 1 clearly represents a dominant controlling factor of surface chemistry related to the particle affinity effect on patterns of the sedimentary OM distribution regardless of OM sources. PC 2 is more likely to reflect the effect of land–source supply, characterized by higher terrestrial compounds contribution, water temperature and turbidity but low salinity of river plume output signal with enhanced impact of PC 2.

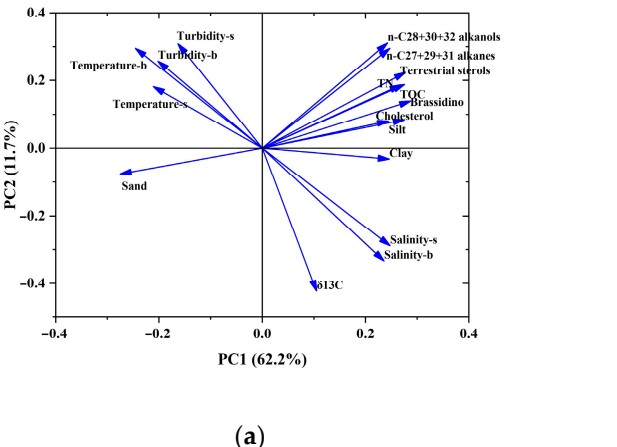
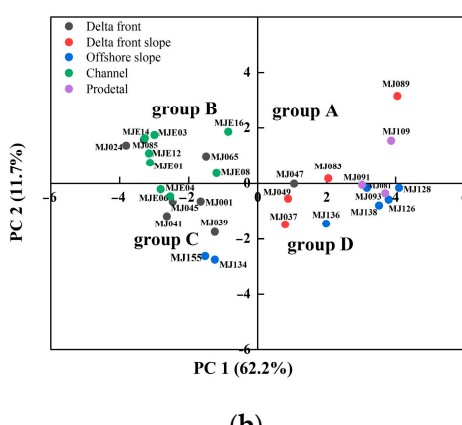

(**a**)　　　　　　　　　　　　　　　　　　　　　　　　　　　　　(**b**)

**Figure 7.** Plot of (**a**) variable loadings and (**b**) sample scores of PC 1 and PC 2 from PCA of samples from surface sediments of the MRE.

## 4. Discussion

### 4.1. Controlled Factors of Spatial Distribution of Sedimentary OM

A sample score plot of the PCA is used to examine relationships between the principal components and the sampling sites (Figure 7b). Most sites show higher scores on PC 1 than on PC 2, which indicates that the distribution of sedimentary OM in the MRE are significantly influenced by the depositional environment commonly regulated by physical processes within the estuary, such as tidal process, fluvial input, resuspension–deposition cycles especially at the turbidity maximum zone and the Zhe-Min Coast current [51–53]. Four distinct groups (A–D) are separated based on their sample scores, with positive or negative scores on PC 1 and PC 2 used to determine the grouping. Samples MJ089 and MJ109, taken from the delta front slope and prodelta, are classified in group A due to their positive scores on both PC 1 and PC 2. This suggests that these samples have a strong affinity and sorption of fine particulate matter to organic components [54], as well as a high accumulation of terrestrial-derived organic matter. The samples in Group B display negative scores on PC 1 but positive scores on PC 2 and are found in the water channel and the delta front near the channel outlet. These areas are characterized by coarser sediments and a greater supply of terrestrial sources. Samples in Group C are distributed throughout the whole estuary and exhibit characteristics of coarser sediment and "negative supply" of terrestrial sources. This suggests that tidal processes have modified the main source of sedimentary OM for this group. The formation and development of TMZ in various estuaries, including those with weak, medium and strong tides, are primarily influenced by estuarine circulation and tidal action. These forces facilitate the flocculation and deposition of metal contaminants and organic matter [55,56]. Based on the in situ observation of water

turbidity at MRE, it was found that TMZ typically occurred at the outlet of the Meihua Channel (Figure 2e,f). Additionally, large gradient variations induced by the salinity front, where saline and freshwater meet (Figure 2c,d), enhance the process of particle flocculation and setting, ultimately leading to accumulation [57,58]. As a result, these river proximal sites are not only the depocenter for terrestrial OM, but also the retained zone for marine particles carried by the tide during high tides. In such tide dominated estuaries, the repeated settling and erosion of particulate matter is caused by strong tidal processes, which affects the entire estuary. As a result, samples falling into group C do not exhibit any specific regional bias. Samples in group D exhibit notable positive scores on PC 1 and fewer negative scores on PC 2. This suggests a high OM associated with fine particles as well as less/more terrestrial/marine derived OM contribution at these sites. Group D samples are mainly found in offshore slope and prodelta, as well as delta front slope (MJ037, MJ047, MJ049). These areas are characterized by high nutrients and low turbidity (Figure 2c–f), which promote the growth of marine phytoplankton and the production of marine OM. Overall, the spatial distribution of sedimentary OM in the MRE is largely determined by the nature of sediment particles. Additionally, sedimentary OM of the sites in the channel, offshore slopes and partial delta front areas are significantly influenced by supply source and tide dynamic forcing.

Higher concentrations of terrigenous *n*-alkanes, straight-chain alkanols and sterols are predominantly found at the delta front slope and prodelta, with the highest values at the MJ89 site in the central of delta front slope (Figure 6a–c), suggesting that the delta front slope and prodelta serve as the primary accumulation area for terrestrial derived OM. The terrigenous materials transported by the river into the delta front via the Meihua Channel are not well preserved at the delta front due to the coarse sandy deposition (Figure 4), causing the terrigenous OM to be carried further eastward to the delta front slope and prodelta. It is at this location where the main depocenter of fine particles as well as OM of MRE can be found (Figures 3b,c and 5a,b). Terrestrial derived OM display a trend of "low-high-low" along the fine particle dispersal belt located at the delta front slope and prodelta from the north to south. The central region of the delta front slope and prodelta exhibit a significantly high accumulation of terrestrial OM. In both the southern and northern regions, the content of terrestrial OM is lower compared to the central part (Figure 6a–c) due to the greater distance from the river source region. Although distal terrestrial material from the Yangtze River can be transported to the north of the delta front slope by the southward Zhe-Min coastal current along the coast of Zhejiang and Fujian [59,60], the terrestrial OM content in this area is low. This is due to the long-distance transport and continuous diagenesis that diluted terrestrial OM delivered by the fluvial source [4,61]. Therefore, the accumulation of terrestrial OM in the MRE is determined not only by the features of the local fluvial source but also by the influence of the distal fluvial source.

Dissolved oxygen is essential for aquatic organisms and serves as an indicator of organic matter decomposition in water [62–64]. The channel of MRE had a low DO saturation and high turbidity (Figure 2e–h), which lead to limited in-situ OM production. In other parts of the subaqueous delta, the presence of higher DO levels in surface water supported the production of OM through phytoplankton photosynthesis. However, the accumulated OM amount in various regions is ultimately determined by differences in depositional conditions. The offshore slope is characterized by a high DO saturation in the surface, but a low DO saturation near the bottom (Figure 2g,h), which creates ideal conditions for the OM deposition [65]. As a result, there was a greater accumulation of sedimentary OM in the northern and southern regions of the offshore slope, as shown in Figure 6d. The spatial distribution of OM in the surface sediments of MRE is primarily controlled by the sedimentary environment and sediment particle size, as well as the supply of a different source. Additionally, alterations in the aquatic environment, such as the influence of DO levels on the decomposition of organic matter, as well as modifications to the sedimentary environment caused by runoff and tide, are significant factors that must be considered.

### 4.2. Source Apportionments of Sedimentary OM in the MRE

#### 4.2.1. TMBR Indices of Terrestrial vs. Marine OM Input Signals

TMBR indices are the ratio of terrestrial plants and marine phytoplankton biomarkers. The ratio can eliminate the influence of sediment size and preferably indicate the source of OM. The TMBRol values range from 0.24 to 1, with an average of 0.63. Samples from the channel and the center of delta front exhibit higher TMBRol (Figure 8a), indicating a greater contribution of OM from terrestrial sources. In the prodelta and offshore slope, TMBRol values decreased in a seaward direction. The minimum is observed at MJ134, which is below 0.3, indicating its sedimentary OM was dominated by marine sources. The OM source apportionment estimated by TMBR was similar to that of TMBRol (Figure 8a,b), with a range of 0.23–1 and an average value of 0.38. Both indices demonstrate a trend of decreased proportion of terrestrial OM accumulation from the estuary to the continental shelf.

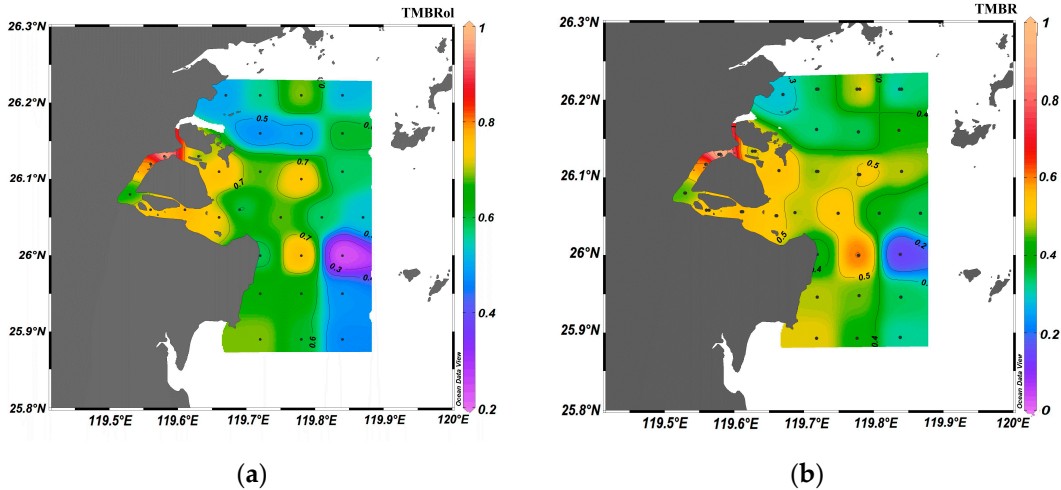

**Figure 8.** Spatial distributions of (**a**) TMBRol and (**b**) TMBR value in surface sediments of MRE.

The distribution of three types of biomarkers, alkanes, alkanols and sterols, indicate that the OM in the surface sediments of the MRE shows complexed signals of terrestrial and marine sources.

#### 4.2.2. Bulk OM Indices of Marine, Delta and Terrestrial Sourced OM Input Signals

The bulk C/N values found in the surface sediments of MRE (ranging from 2.9 to 10.2) overlap with the expected values from the mixture of marine phytoplankton (C/N = 5–9) [12]. Delta OM was originated from the 0–5 cm sediment of Shanyutan and Bianfuzhou wetlands at Meihua Channel of MRE (C/N = 8.76–9.82) [66] and fluvial particulate OM exported by Minjiang River downstream (C/N = 9.42–12.17, avg. 11.28) [67]. In this study, the C/N values of stations at MJE12 and MJE14 were lower than six, while the C/N values of most other stations ranged between seven and ten (Figure 5c). These values are slightly lower than those reported in previous studies. The results indicate that both terrigenous and marine sources significantly contributed to the surface OM in the study area.

According to previous studies on the $\delta^{13}C$ values of OM from various sources, typical $\delta^{13}C$ values of terrestrial C3 plants range from −34‰ to −19‰, mostly around −27‰ [68,69]. Typical $\delta^{13}C$ values of marine phytoplankton range from −19‰ to −22‰; the average is −20.5‰ [70]. In this study, the $\delta^{13}C$ values observed in the surface sediment of MRE ranged from −26.8‰ to −19.9‰ (avg. −23.4‰), indicating a mixture of both marine and terrestrial OM accumulation. The $\delta^{13}C$ values in channel sediments (−25.3‰ to −23.22‰) are lighter than those outside the channel (Figure 5d), showing obvious signal of terrestrial OM accumulation. The $\delta^{13}C$ value of the offshore slope on the near marine side are heavier than that found in the underwater delta area (Figure 5d), indicating the signal of marine OM

accumulation, which is consistent with the outcome derived from biomarker ratios indices such as TMBRol and TMBR (Figure 8). The minimum $\delta^{13}C$ value is observed at the MJ085 from the delta front (Figure 5d), suggesting its high terrestrial derived OM accumulation. This finding aligns with the TMBR indices results (Figure 8). Hui et al. (2020) [71] observed the $\delta^{13}C$ values of suspended particulates in the MRE. The range of values observed was from $-25.2‰$ to $-21.8‰$. The $\delta^{13}C$ values were heavier in the delta front than those in the channel entrance. The variation range and spatial distribution of $\delta^{13}C$ values were basically consistent with those observed in this study.

### 4.2.3. Quantitative Source Analysis of Sedimentary OM

A plot of bulk OC $\delta^{13}C$ vs. C/N was used to constrain the OM sources buried in the MRE (Figure 9). Most samples fall along the mixing line between a marine source endmember and a typical C3 terrestrial endmember. Samples located in the north (MJ037, MJ083) and southeast (MJ049) part of the delta front slope and the southeastern delta front (MJ045) fall above the mixing line. This suggests the presence of another source of $^{13}C$-enriched or high C/N sources input, possibly from wetland derived OM distributed at the proximity of MRE that contain significant proportions of C4 fresh plant or partially degraded soil signal [72,73]. Samples falling below the mixing line were mainly collected from channels with sandy sediments and they exhibit $^{13}C$-depleted or low C/N signals compared to other samples. Such lower C/N signals of bulk OM have been usually observed at estuaries and coastal environments, resulting from adsorption of inorganic nitrogen with sediments due to a large amount of eutrophic fluvial input [70,74,75]. As Figure 5 shows, the extremely low C/N signals have been also observed at the delta plain proximity near the river mouth. Based on the above discussions, a three-end-member mixing model based on a certain degree of variable ranges of $\delta^{13}C$ and C/N (Table 1) has been applied to quantify relative fractional proportions of three different components: marine biomass ($f_M$), terrigenous input through the Minjiang River ($f_R$) and deltaic wetland ($f_D$) OM in the surface sediment of MRE. Calculation methods such as Formula (3), the values of end members are based on the data of C/N and $\delta^{13}C$ from previous investigations of the Minjiang River basin and estuary. Considering the spread end-member parameter values, we select the typical range of $\delta^{13}C_i$ and $C/N_i$ for specific OM sources to constrain the solutions of this isotopic mass balance model between 1 and 0 via MC simulations [76,77]. As Table 1 shows, river sourced OM in the three-end-member mixing model assumed a range of $C/N_R$ from $12.4 \pm 1.2$ and $\delta^{13}C_R$ values from $-27.0 \pm 2.5‰$ as those of soil sediment or suspended particulate matter in the downstream of the Minjiang River [78,79]. Marine sourced OM assumed a range of $C/N_M$ from $5.0 \pm 2.0$ and $\delta^{13}C_M$ values from $-20.0 \pm 2.0‰$, which are typical values of marine phytoplankton [70,80]. For delta sourced OM, the spartina alterniflora of the MRE has invaded and grown rapidly in recent years, resulting in the formation of a large area of spartina alterniflora salt marshes [81]; therefore, the end member values of the deltaic wetland assumed a range of $C/N_D$ from $10.5 \pm 1$ and $\delta^{13}C_D$ values from $-14 \pm 1.0‰$, which are similar to those of the spartina alterniflora in the MRE [73,81,82].

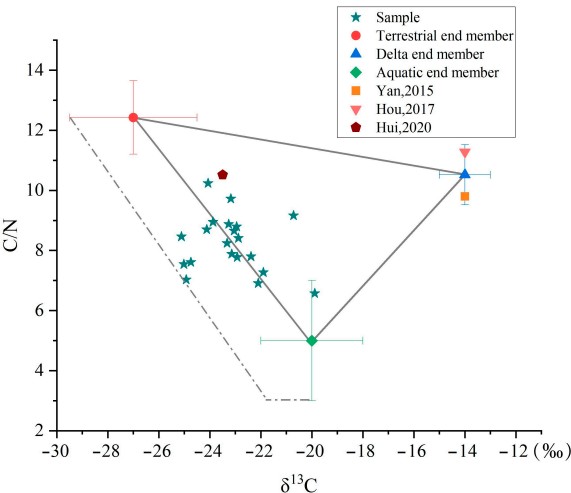

**Figure 9.** Distribution of three end members and samples of MC Model. The values of square point is cited from Yan et al. (2015) [66], the values of inverted triangle point is cited from Hou (2017) [67], the values of pentagonal point cited from Hui et al. (2020) [71].

**Table 1.** End member values of MC Model.

| Sources | Parameter | Values | References |
|---|---|---|---|
| River | C/N | $12.4 \pm 1.2$ | Hu et al. (2016) [78] |
| | $\delta^{13}C / ‰$ | $-27.0 \pm 2.5$ | Fry and Sherr (1989) [79] |
| Delta | C/N | $10.5 \pm 1.0$ | Jin et al. (2016) [81]; LI et al. (2016) [82] |
| | $\delta^{13}C / ‰$ | $-14.0 \pm 1.0$ | Shang et al. (2009) [73] |
| Marine | C/N | $5.0 \pm 2.0$ | Stein (1991) [80] |
| | $\delta^{13}C / ‰$ | $-20.0 \pm 2.0$ | Hedges et al. (1997) [70] |

According to the results of the MC model, the riverine ($f_R$), marine ($f_M$) and deltaic wetland ($f_D$) OM fractional contribution range from $15 \pm 9\%$–$61 \pm 11\%$ (avg. $40 \pm 10\%$), $24 \pm 12\%$–$64 \pm 12\%$ (avg. $48 \pm 10\%$) and $5 \pm 5\%$–$20 \pm 11\%$ (avg. $12 \pm 4\%$), respectively (Table 2). The high fractional contribution (>50%) of riverine OM accumulation was detected in the sites from the Changmen Channel, north of the Culu Island and delta front of the Meihua Channel outlet, while the low riverine fractional contribution appeared at the offshore slope (Figure 10a). The spatial variations were consistent with those evaluated by biomarker ratio indices such as TMBRol and TMBR (Figure 8). The results indicate that a river-proximal depocenter and river-distal depocenter of terrestrial OM transported by the Minjiang River are located at the Changmen Channel and southeastern delta front region, respectively (Figures 8 and 10a). The spatial distribution of marine OM fractional contribution in the surface sediment of the MRE is generally opposite to that of riverine OM (Figure 10a,b). The highest proportions of marine derived OM accumulation were found in the offshore region far from the river mouth, and at the northern extremity of the study area, which is located outside of the Ao'jiang Estuary. Futhermore, a significant contribution of marine-derived OM has been observed at the Meihua Channel (Figure 10b), indicating that the Meihua Channel is the main tidal inlet for the marine derived materials input [83]. For the strong tidal estuary, the channels region can accumulate terrestrial OM from the fluvial supply as well as marine derived OM input via tidal processes. Similarly, the TMBRol and TMBR indices indicate a great marine derived OM accumulation signal at the Meihua Channel compared to the Changmen Channel (Figure 8), suggesting a difference in terrestrial organic matter proportions between the two locations. The contribution of local delta derived OM only accounts for a minor portion of the total OM accumulation in the MRE (avg. $12 \pm 4\%$). Generally, the relative contribution of delta derived OM in total sedimentary OM of the MRE shows a high value at distal sites off the river mouth

and a low value at proximal sites near the river side. Goñi et al. (1998) [74] proposed that hydrodynamic sorting caused by the resuspension and cross-estuary transport of particles may serve to physically segregate distinct pools of terrestrial OM. The result of this process is the preferential transport of C4 grassy wetland derived OM from the delta closely associated with the finer-grained mineral load (clays) to exterior regions of the MRE. In contrast, coarser and denser OM containing water-logged C3 vascular plant detritus (likely derived from the abundant upland and lowland forests) is retained in the channels and the inner delta front along with silt and sand-sized mineral particles. In addition, the highest proportions mainly occurred at the northern part of the study area (Figure 10c), which was consistent with distributions of typical spartina alterniflora dominant wetlands at the MRE [72]; therefore, the delta source contribution is relatively obvious.

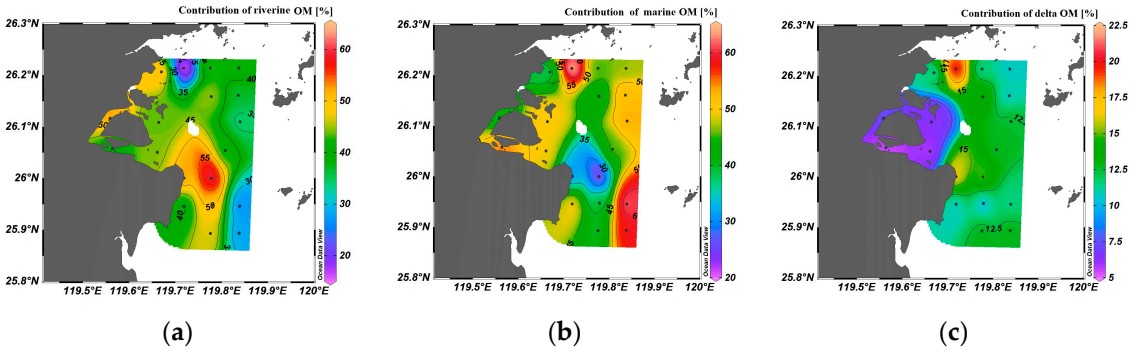

**Figure 10.** Spatial distributions of contribution of (**a**) riverine OM, (**b**) marine OM and (**c**) delta derived OM in surface sediments of MRE.

Quantitative estimation for OM source apportionment based on multiple organic geochemical indicators in different aquatic environments using an MC model has been applied in several estuaries in southeastern China (Table 2). A previous study estimated that the fractional contribution of sedimentary OM from the surface sediments within the Yangtze River Estuary and the inner shelf along the southeast coast of China accounted for $18 \pm 5\%$ from riverine OM, $27 \pm 5\%$ from delta derived OM and $56 \pm 6\%$ from marine OM, respectively [76], and the fractional contribution of sedimentary OM from the surface sediments within the Pearl River Estuary and the northern South China Sea accounted for $44 \pm 6\%$ from riverine OM, $22 \pm 10\%$ from marine OM, $34 \pm 4\%$ from delta derived OM, respectively [84]. Bao et al. (2013) [85] investigated the source and compositions of sedimentary OM with the growth of a large amount of mangrove in the Wenchang Estuary located at the southern Chinese coast based on parameters of bulk OC $\delta^{13}$C and the lignin contents and found that the contribution of local mangrove OM into the interior of the estuary was significant and predominated ($59 \pm 17\%$, Table 2). Cheng et al. (2021) [86] quantified different sourced sedimentary OM in the Jiulong River Estuary based on $\delta^{13}$C and C/N, with contributions of $40 \pm 9\%$, $27 \pm 6\%$ and $33 \pm 7\%$ from riverine, upland soil and marine sources, respectively. In conclusion, according to the fractional contribution of sedimentary OM from diverse sources in the river–estuary sediment continuum in southeast China (Table 2), the marine input signal is pronounced in estuaries with a strong tidal impact (mean tidal range > 4 m) (e.g., MRE), and marine OM imported from the outer sea accounts for half of the whole sedimentary OM in the estuarine. In mid-small mountainous estuaries (e.g., MRE and Jiulong River Estuary), terrestrial OM is rapidly buried after entering the estuaries, exhibiting a significant riverine buried OM signal and a less delta-derived OM signal; smaller river–mangrove estuaries in southeast China (e.g., Wenchang Estuary) are dominated by organic matter derived from local deltaic wetland sources.

**Table 2.** Contribution from different source OM buried in estuaries of southeast China calculated by MC Model.

| Location | Gradient of the River | Mean Tidal Range (m) | Source from River (or Soil) | Source from Marine | Source from Estuary Delta (or Marsh Plant) | References |
|---|---|---|---|---|---|---|
| Minjiang River Estuary | 5.0‰ | 4.11 | 40 ± 10% | 48 ± 10% | 12 ± 4% | (this study) |
| Jiulong River Estuary | 6.2‰ | 3.37 | 67 ± 11% | 33 ± 7% | Minor | Cheng et al. (2021) [86] |
| Yangtze Estuary | 0.7‰ | 2.70 | 18 ± 5% | 56 ± 6% | 27 ± 5% | Li et al. (2014) [76] |
| Pearl River Estuary | 0.4‰ | 1.31 | 44 ± 6% | 22 ± 10% | 34 ± 4% | Li et al. (2017) [84] |
| Wenchang Estuary | 1.7‰ | 0.75 | 12 ± 7% | 29 ± 21% | 59 ± 17% | Bao et al. (2013) [85] |

Note: The mean tidal range of estuaries are cited from Chinese Harbours and Embayments (Volume 14): Important Estuaries [87]; gradient of Minjiang River is cited from Zhang (2000) [88]; gradient of Pearl River is cited from Luo et al. (1985) [89].

Overall, this study shows a more comprehensive view of OM accumulation in different types of estuarine systems in southeast China. In large rivers and estuaries, local deltaic wetland is a significant source of sedimentary carbon accumulation. However, in tide dominated small and medium-sized estuaries, the contribution of OM accumulation is largely dependent on riverine and marine delivery with the local delta playing a less significant role. In regard to the spatial distribution of buried OM, the physical process plays more roles on OM composition and fates in strong tidal estuaries. Compared to large scale broad shelf estuaries, terrestrial OM is derived and buried more rapidly in small mountainous rivers and estuaries; on the other hand, tidal processes exert a more prominent effect on the redistribution of sedimentary OM in the estuary.

### 4.2.4. Implications for Organic Carbon Cycling and Budgets in MRE

Based on the quantitative estimations, the sedimentary OM fractional contribution from various sources in the surface sediment of MRE was determined. This allowed for the calculation of the deposition amount of particulate organic carbon (POC) from river, marine and delta marsh in different sedimentary zones, including sandy and muddy sediment. Table 3 shows the mass accumulation rate of sediment and TOC in the MRE. The TOC accumulation rate in the muddy zone of MRE is higher than that of the sandy zone. This is consistent with the fact that fine particles have a stronger affinity and sorption to organic matter compared to coarse particles. Additionally, the sediment accumulation rate is higher in the prodelta than in the delta front and offshore slope [90–93]. Compared with the Pearl River Estuary (13.02 mg cm$^{-2}$ yr$^{-1}$) [84], the MRE has a comparative potential for organic carbon burial (9.40 ± 3.54 mg cm$^{-2}$ yr$^{-1}$ in muddy sediment). The terrestrial OC accumulation in the MRE (3.39 ± 1.83 mg cm$^{-2}$ yr$^{-1}$ in muddy sediment) is slightly lower than that found in the Pearl River Estuary (7.80 ± 0.77 mg cm$^{-2}$ yr$^{-1}$), while the marine OC accumulation in the MRE (3.18 ± 0.68 mg cm$^{-2}$ yr$^{-1}$ in muddy sediment) is significantly higher compared to the Taiwan Strait (0.83 ± 0.17 mg cm$^{-2}$ yr$^{-1}$) (Tao et al. unpublished). Those results indicate that mid-small mountainous estuaries in southeast China are an important area for sedimentary carbon sink.

The carbon cycle in the MRE and the deposition of organic carbon in different sedimentary zones of the subaqueous delta are depicted in Figure 11. Previous investigations have shown that the flux of riverine POC input from rivers into the MRE is 6.0 × 10$^{10}$ g C yr$^{-1}$ [94]. Additionally, there is a net release flux of 7.74 × 10$^{10}$ g C yr$^{-1}$ to the atmosphere through sea–air CO$_2$ exchange in the MRE region [95]. The riverine POC undergoes a range of biogeochemical processes, including microbial utilization and organic matter decomposition [96–99], as well as physicochemical processes such as adsorption, desorption, dissolution and settling [12,13,100] when it enters the complex estuarine environment. Our assessment indicates only approximately 28% of the POC from fluvial input

is deposited in the estuary ($(1.69 \pm 0.54) \times 10^{10}$ g C yr$^{-1}$), with the majority being found in the prodelta muddy sediment (Figure 11). The missing POC can be either suspended in the water or converted into dissolved organic carbon (DOC) or $CO_2$ through microbial utilization, which can either be stored permanently as recalcitrant DOC or released into the atmosphere or transported laterally to the sea. The organic carbon formed in the adjacent marginal sea can be transported to the estuary by tides, where it plays a crucial role in the carbon cycle through biological or physical pumps. Our estimation suggests that the amount of marine organic carbon deposited in the MRE is comparable to that of riverine deposited POC with $(1.69 \pm 0.54) \times 10^{10}$ g C yr$^{-1}$ of the riverine source and $(2.06 \pm 0.54) \times 10^{10}$ g C yr$^{-1}$ of the marine source, respectively. In a hypothetical extreme scenario where all POC input from the river, excluding the amount deposited in the estuary, is released into the atmosphere as $CO_2$ at the sea–air interface without any other retention and conversion in the aquatic environment or transportation to the sea. As a $CO_2$ source, the MRE releases maximum $(4.31 \pm 0.54) \times 10^{10}$ g C yr$^{-1}$ of terrestrial OM degraded $CO_2$ and at least $(3.43 \pm 0.54) \times 10^{10}$ g C yr$^{-1}$ of marine sourced OM degraded $CO_2$ into the atmosphere. Therefore, it is inferred that only a small percentage (approximately 30% of riverine organic carbon) is deposited in such tidal estuaries. Most organic carbon will either decompose or remain suspended in the aquatic environment, participating in various biochemical or physical processes.

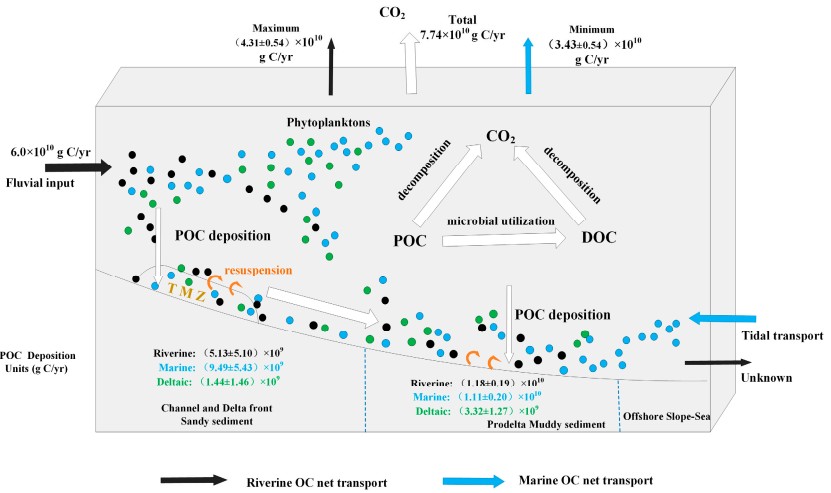

**Figure 11.** Graphical figure of carbon cycles and OC deposition of MRE.

**Table 3.** Mass accumulation rate of sediment and TOC in the MRE and other estuaries.

| Parameters | Estuary | Channel and Delta Front Sandy Sediment | Prodelta Muddy Sediment | Offshore Slope |
|---|---|---|---|---|
| Sediment accumulation rate (g cm$^{-2}$ yr$^{-1}$) | Minjiang River Estuary | $0.26 \pm 0.13$ [90,92] | $0.99 \pm 0.09$ [91] | 0.61 [93] |
| | Pearl River Estuary [84] | 1.3 | | 0.156–0.468 |
| TOC accumulation rate (mg cm$^{-2}$ yr$^{-1}$) | Minjiang Estuary | $0.80 \pm 0.78$ | $9.40 \pm 3.54$ | $3.99 \pm 2.18$ |
| | Pearl River Estuary [84] | 13.02 | | 0.67–2.01 |
| Terrestrial OC accumulation rate (mg cm$^{-2}$ yr$^{-1}$) | Minjiang Estuary | $0.34 \pm 0.34$ | $3.39 \pm 1.83$ | $1.29 \pm 0.91$ |
| | Pearl River Estuary [84] | $7.80 \pm 0.77$ | | 0.24–0.72 |
| Marine OC accumulation rate (mg cm$^{-2}$ yr$^{-1}$) | Minjiang Estuary | $0.36 \pm 0.36$ | $3.18 \pm 0.68$ | $2.18 \pm 1.54$ |
| | Taiwan Strait [101] (unpublished) | $0.83 \pm 0.17$ | | |

## 5. Conclusions

The OM content in the surface sediments of MRE is significantly affected by the source, the carrier, sedimentary environment and both the total organic matter and biomarkers

content are negatively correlated with sediment grain size. The fine sediment adsorbed OM more easily than the coarse sediment. The sediments in the channel, delta front and offshore shelf are mostly coarse sand with low content of OM, and the sediments in the delta front and prodelta are clay with a high content of OM. Tidal processes play a significant role on redistribution of sedimentary OM accumulated in MRE. The bulk OM and biomarker indicators in the MRE sediments demonstrate that the sedimentary OM in this area is mainly imported by the terrestrial source and the marine source (including planktonic and microbial biomass). Using the C/N and $\delta^{13}$C coupling three end member mixed model to estimate the relative contribution of riverine, deltaic and marine source in estuarine sedimentary OM, the calculated results show that the main source of sedimentary OM in the MRE are river and marine derived, with the average fractional contribution rates of $40 \pm 10\%$ and $48 \pm 10\%$, respectively, and that of the deltaic source is lower, with an average fractional contribution of $12 \pm 4\%$. Thus, the OC accumulation contribution in tidal estuaries such as the MRE is largely dependent on riverine and marine delivery in tide dominated small and medium-sized estuaries, whereas the contribution of the local delta is not significant. The MRE has a comparative potential for organic carbon burial as large rivers and estuaries such as the Pearl River Estuary and it exhibits higher accumulation rates for both terrestrial and marine OC. Those finding suggests that mid-small mountainous estuaries play a significant role in the sedimentary carbon sink. Using the MRE as a case study, it was found that approximately 30% of the fluvial input POC is deposited in the estuary in the mid-small mountainous river-wide continental shelf system, with the majority of it being found in muddy sediments. Additionally, in strongly tidal estuaries, the deposition of both terrestrial and marine OC is comparable.

Unfortunately, our mass balance estimation does not accurately reflect the actual OC amount present at different interfaces. To obtain a more precise understanding of carbon cycling processes and exchange patterns in the estuary, additional research is required to examine carbon fluxes at various key interfaces. Furthermore, it is necessary to elucidate the mechanisms of organic carbon transformation within the estuary, particularly at specialized zones such as TMZ and the anoxic area.

**Author Contributions:** Conceptualization, S.T. and X.Y.; data curation, S.W.; formal analysis, S.W.; investigation, X.Y. and A.W.; methodology, S.T.; project administration, X.Y.; resources, A.W.; software, Z.L. and W.Z.; validation, C.R., H.L. (Haiqi Li) and Y.Y.; visualization, H.L. (Haoshen Liang), Y.Y. and W.Z.; writing—original draft, S.W.; writing—review and editing, S.T., A.W. and J.T.L. All authors have read and agreed to the published version of the manuscript.

**Funding:** This study was supported by the Scientific Research Foundation of Third Institute of Oceanography, MNR (Grant No. 2017015 and 2019018); Natural Science Foundation of Fujian Province (Grant No. 2020J05076); National Natural Science Foundation of China (Grant No. 41961144022, 41776099 and 41506089); Innovation Group Project of Southern Marine Science and Engineering Guangdong Laboratory (Zhuhai) (Grant No. 311021004).

**Data Availability Statement:** The original contributions presented in the study are included in the article. Further inquiries can be directed to the corresponding authors.

**Acknowledgments:** We would like to thank Shuren Huang, Yongze Yu, Haihuang Chen, Sitian Huang and Sanshan Liu for sampling; thank Yijun Lan and Wenjuan Su at the TIO for help with organic geochemical analyses.

**Conflicts of Interest:** The authors declare that they have no known competing financial interest or personal relationships that could have appeared to influence the work reported in this paper.

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
