# Peer review of "Characteristics of Sedimentary Organic Matter in Tidal Estuaries: A Case Study from the Minjiang River Estuary"

_water, doi:10.3390/w15091682_

Round 1
Reviewer 1 Report
The article by Wu et al. deals with the sedimentary organic matter in the tidal estuary of the Min River. The topic is very interesting and this paper falls within the scope of the journal Water. Besides, it could be a nice contribution.
1. Usually, the modelling of the endmembers is set with a constant or a constrained value, such as 1 or 100%. The authors have given average values of more than 100% in the abstract and in some other parts of the paper. Can they explain this or was this an error in the calculation?
2. Why did the authors decide to remove organic matter before calculating the particle size distribution, and would the results have been different if they had not done so?
3. This type of unsupervised analysis (PCA) was not so clear in the selection of sample patterns. It is not clear from the authors' explanation how they chose samples in each group. We can only believe their explanations. Perhaps it would be better to add the sample names and multidimensional scaling, which corresponds to PCA for Euclidean distance. Arrows are usually used for variables and not for samples and are therefore confusing. I suggest the authors to remove the arrows. Perhaps K-means clustering is also a method that can be chosen.
------Minor comments-----
Line 28: No semicolon.
Line 33: How can mixing models estimate be greater than 100%?
Line 117: Space please and so on…
Line 195-206: Please add software for mixing modelling or it was calculated simply in excel?
Line 275: And which one is it?
------Tables and Figures------
No comments.
Author Response
Dear Editor:
We want to thank you, Associate Editor and the three reviewers for the most detailed and constructive comments-suggestions. We have been working very hard to answer all the comments and requests. You will see from the separate lists of responses (in a separate word file) that we have been very conscientious in meeting the requests for changes and we have done virtually everything asked for, including rewriting introduction, polishing language, making new figures and modifying data and error propagation. We are sorry that it took us some time to finish the revision, but we believe the paper is considerably improved and you will be satisfied with the revised MS. Please see the respond in the attachment
Sincerely,
Shuqin Tao

Reviewer 2 Report
In current study author characterize the organic matter in Min River Estuary sediments. Although the results and methodology portion are presented in good manner. But lacking the significance of the study. Objectives of the study were unclear that what authors want to study. Why author needs to study the OM in estuaries? Author must describe the significance of the study that how this study helpful in sediment management. Dissolve oxygen is an important parameter which impacts the aquatic life in water bodies. Organic matter degradation strongly effects the concentration of dissolve oxygen (DO) in water. Author should develop a correlation between OM and DO concentrations. In abstract add some important outcomes of the study. Support your results with latest literature, most of the references are very old. Add graphical figure of carbon cycles of MRE on the bases of your results.
Author Response
Dear Editor:
We want to thank you, Associate Editor and the three reviewers for the most detailed and constructive comments-suggestions. We have been working very hard to answer all the comments and requests. You will see from the separate lists of responses (in a separate word file) that we have been very conscientious in meeting the requests for changes and we have done virtually everything asked for, including rewriting introduction, polishing language, making new figures and modifying data and error propagation. We are sorry that it took us some time to finish the revision, but we believe the paper is considerably improved and you will be satisfied with the revised MS.
Sincerely,
Shuqin Tao

Reviewer 3 Report
This manuscript quantifies organic matter and biomarkers in surface sediments within the Min River Estuary and applies a 3-end member mixing model to estimate contributions from terrestrial, delta, and marine sources. The methods and results are well described and reported and are placed into the broader context of similar studies in the discussion. My main suggestion is to better connect the introduction and discussion sections. For example, natural vs. anthropogenic sources of OM are discussed in the introduction (lines 69-78) and how this can be used to link human and natural activities to environmental changes in the estuary. However, this point is not revisited in this manuscript (outside a brief mention of coprostanol being derived from human sewage). I would suggest removing this from the introduction since the focus of this manuscript is on surface sediments and therefore is not appropriate for evaluating changes over time. Another recommendation I have is to reevaluate using the term “buried” when describing the OM in surface sediments. There are a few instances of this word throughout the manuscript (i.e., line 477), but I do not think this is the appropriate word when describing surface sediments that are actively being reworked/undergoing sediment diagenesis. This sediment has the potential to be buried (and thus the OM preserved); however, I do not think OM burial can be assessed without collecting a longer core.
Please see below for some specific comments.
Line 46 – Gt = 10^15 g
Lines 181-183 – I don’t think you need so much description for PCA since you have a citation.
Describe the interpolation methods (linear/inverse distance weighting, Kriging) used to describe spatial variability in environmental parameters/grain size/etc.
Line 260 – Missing label STY in text
Lines 417-421 – I find this sentence hard to follow
Line 422 – what is the center vs. side in “high in the center low on the side” referring to?
Figure 9 – I would change the “Aquatic end member” legend entry to “Marine end member”
Author Response

(The authors gave the same response as above.)

Round 2
Reviewer 1 Report
The paper has been considerably improved and can be accepted as it stands.
Reviewer 2 Report
Before accepting the manuscript formating issues should adress, author must go through the grammaticaol mistakes which must resolved.